# GLoMo: Global-Local Modal Fusion for Multimodal Sentiment Analysis

## ABSTRACT

Multimodal Sentiment Analysis (MSA) has witnessed remarkable progress and gained increasing attention in recent decades, thanks to the advancements in deep learning. However, current MSA methodologies primarily rely on global representation extracted from different modalities, such as the mean of *all* token representations, to construct sophisticated fusion networks. These approaches often overlook the valuable details present in local representations, which consist of fused representations of consecutive *several* tokens. Additionally, the integration of multiple local representations and the fusion of local and global information present significant challenges. To address these limitations, we propose the Global-Local Modal (GLoMo) Fusion framework. This framework comprises two essential components: (i) modality-specific mixture of experts layers that integrate diverse local representations within each modality, and (ii) a global-guided fusion module that effectively combine global and local representations. The former component leverages specialized expert networks to automatically select and integrate crucial local representations from each modality, while the latter ensures the preservation of global information during the fusion process. We extensively evaluate GLoMo on various datasets, encompassing tasks in multimodal sentiment analysis, multimodal humor detection, and multimodal emotion recognition. Empirical results demonstrate that GLoMo outperforms existing state-of-the-art models, validating the effectiveness of our proposed framework.

## CCS CONCEPTS

• **Computing methodologies** → **Neural networks**; • **Information systems** → Multimedia information systems; • **Sentiment analysis**;

## KEYWORDS

multimodal sentiment analysis; multimodal fusion; multimodal representation learning

## 1 INTRODUCTION

Multimodal Sentiment Analysis (MSA), which aims to infer human emotions by leveraging signals from various modalities [11], has witnessed remarkable progress and gained increasing attention,

*ACM MM, 2024, Melbourne, Australia*

© 2024 Copyright held by the owner/author(s). Publication rights licensed to ACM.
ACM ISBN 978-x-xxxx-xxxx-x/YY/MM
https://doi.org/10.1145/nnnnnnn.nnnnnnn

thanks to the surge of videos and advancements in deep learning [2, 26, 43, 58]. Most current research is focused on developing complex fusion networks that integrate multimodal representations from heterogeneous sources. These include models that fuse representations across modalities using MLP and attention-based mechanisms [25, 46], as well as methods that employ LSTM [44], or transformer encoder [49] techniques.

Although these methods have been quite successful, they either focus only on the global representation of each modality, such as pooling or averaging all tokens for audio and image modalities [51, 64], using BERT's [*CLS*] token [20] to represent the entire text modality [15, 41], or they concentrate solely on token-level fusion [24, 55, 59]. For instance, representations of each token from various modalities are merged and then averaged [7, 59]. These approaches overlook the local representations, which consist of fused representations of consecutive several tokens. Local representations contain a wealth of detailed information, such as the nuanced changes in mouth movements in videos, which can significantly enhance sentiment analysis and effectively capture certain details [18, 31]. Furthermore, how to integrate multiple local representations and how to better fuse local representations with global information, a 'few-to-many tokens' fusion, remains an open question.

To tackle these challenges, we propose the Global-Local Modal Fusion (GLoMo) framework for multimodal sentiment analysis. GLoMo is characterized by two salient attributes. Firstly, it can autonomously fuse multiple local representations to derive the most suitable local representations for each modality. This is achieved through a modality-specific mixture of experts (MoEs) layer [45], which comprises various experts that focus on different aspects of the local presentations. It automatically integrate the outputs from the most relevant experts. Secondly, considering the 'few-to-many token' relationship between local and global representations, it is necessary to maintain the dominant influence of global representations during the fusion process. In other words, modalities that are dominant in global representations should also be dominant in local representations, to keep the consistency. GLoMo integrates the fusion of local representations into global ones, ensuring that global context is preserved while local details are assimilated. Specifically, we have designed a global-guided fusion module that initially merges local and global representations within each modality, then blends all local and global representations to derive attention weights. The attention weights derived from the global feature fusion of the three modalities, are used as mixup weights to merge the modality-specific representations, thus safeguarding the significance of the original modalities in the composite representation.

The key contributions of our work are as follows:

- We introduce the GLoMo, a global-local modal fusion framework for multimodal sentiment analysis, which autonomously fuses local representations using modality-specific mixture

of experts layers, enhancing the model's ability to capture nuanced sentiment cues from different modalities.

- We introduce an innovative global-guided fusion module adept at navigating the 'few-to-many token' relationship between local and global representations, thereby maintaining the primacy of global representations throughout the fusion process.
- Extensive experiments on various datasets demonstrate the superior performance of the GLoMo, with ablation studies validating the effectiveness of the GLoMo's components.

## 2 RELATED WORK

### 2.1 Multimodal Sentiment Analysis

Multimodal sentiment analysis aims to leverage heterogeneous data sources, such as audio, image frames, and textual information from videos, to assess the emotional state and intensity of the individuals depicted [11, 58]. Existing methods can be generally categorized based on the granularity of representations used from each modality into two types: utterance-level and inter-utterance contextual approaches. The former primarily leverages the global representation of modalities, which is typically obtained through averaging or pooling all tokens [64], capturing the final token from a temporal convolution [36, 40], or employing methods such as LSTM or BERT to acquire an overall representation for each modality [15, 56]. On the other hand, inter-utterance contextual methods emphasize the exploration of relationships between tokens across different modalities [24, 55]. For instance, GME-LSTM [7] and DFG [59] perform fusion across tokens from each modality, whereas RAVEB [24] employs an RNN-based approach to capture representations between modalities. ScaleVLAD [31] concentrates on fusing local representations of each modality at various granularities and optimizes them using supervised clustering algorithms.

The aforementioned studies highlight the significant roles of both global representations and local token representations in multimodal sentiment analysis. However, each of these works has focused exclusively on one type of representation. Our work is an integrated approach that considers both global and local representations simultaneously, aiming to harness the combined strengths of both. Moreover, fusing all token representations can be time-consuming and may introduce redundant and noisy data, as noted in [64]. To address this issue, we incorporate a MoEs layer that employs multiple experts to concentrate on different local representations [45], thereby reducing computational overhead and potential errors.

### 2.2 Multimodal Representation Learning

Due to the heterogeneity of data sources, existing multimodal affective computing approaches can be categorized into three main types based on different fusion strategies. The first type involves designing various networks to directly fuse representations from each modality, including techniques such as low-rank order tensor fusion [29, 57], high-order polynomial fusion [19], gated fusion [1, 50], mlp fusion [46], fusion with multimodal transformer encoder [12, 44] and complex fusion [5, 17]. The second type aligns the modalities before fusing them, employing methods like MULT [49] and AcFormer [64] that use transformers to align any two

modalities, or HYCON [40] and MCL [36] which utilize supervised contrastive alignment of corresponding modalities, followed by fusion strategies as in the first type. The third type deconstructs the representations of each modality into shared and unique information. For instance, MISA [15] projects the representation of each modality into modality-specific and modality-variant subspaces, while FACTORCL [23] decomposes the representations into task-relevant shared and unique representations, and PID [22] extends this decomposition to include unique, redundant, and synergistic multimodal information.

Our method bears some resemblance to the third category, as it also partitions modalities. However, unlike the aforementioned approaches which further partition based on the same granularity of global representations [15, 23, 33], our focus is on leveraging both global and local representations of different granularities. Moreover, a distinctive aspect of our work is the role that local and global representations play in the overall classification of information. Our proposed global-guided fusion module is designed to use local representations as an auxiliary to the dominant global representations.

## 3 APPROACH

### 3.1 Task Setup

Multimodal sentiment analysis aims to predict the sentiment intensity or emotion category of the given utterances, which involves information across texts ($t$), audios ($a$) and videos ($v$) [42]. To facilitate this analysis, we represent the sequences from each modality as $U_m \in \mathbb{R}^{T_m \times d_m}$, where $m \in \{t, a, v\}$ denotes the modality, and $d_m$ and $T_m$ represent the respective dimensionality and sequence length, respectively.

### 3.2 GLoMo

**Model Overview.** The diagram of GLoMo is illustrated in Fig. 1. It consists of three modules, namely unimodal coding module, modality-specific MoEs module and global-guided fusion module. The detailed introduction of each module can be found in following subsections.

### 3.3 Unimodal Coding Module

Unimodal coding module aims to extract the global and local representations of the each modality in the given utterance. To be consistent with prior research [15, 36, 40, 56], we adopt the similar way to extract the global representations. For global text representations $X_t^g$, we take the representation of the $[CLS]$ token of the last layer in BERT [20], denoted as $CLS^{-1}$, followed by a feedforward layer to project the dimension into a hidden dimension $d$, as delineated in the following expression:

$$X_t^g = \text{FF}\left(CLS^{-1}\left(U_t\right)\right) \tag{1}$$

To capture the local representations of the textual modality, we concatenate the tokens from the last two layers of BERT, denoted as $[BERT^{-2}, BERT^{-1}]$, and feed them into a temporal convolutional (Conv1D) layer to obtain contextual information for each token. Subsequently, the extracted representations are subjected to adaptivemaxpooling layer to acquire $n$ local representations. These

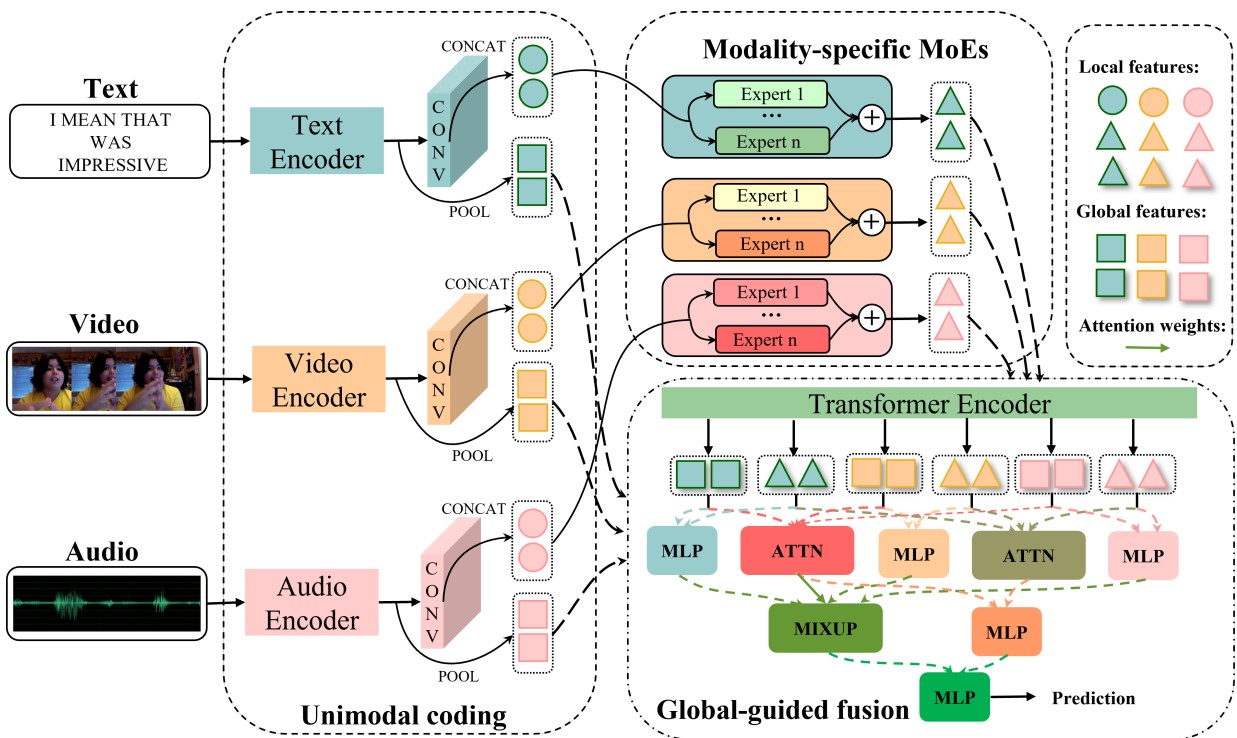

**Figure 1: The diagram of the GLoMo. The text, video and audio representations are firstly processed by modality-specific encoders to get the global representations, followed by modality-specific MoEs layers to get the local representation of each modality. The global and local representations of the three modalities are then fed into the global-guided fusion module for prediction.**

representations are then concatenated. For simplicity, we denote this concatenated vector as $X_t^l$:

$$X_t^l = \text{CONCAT}\left(\text{AdaMaxPool}_n\left(\text{Conv1D}\left([BERT^{-1}, BERT^{-2}]\right)\right)\right) \quad (2)$$

For acquiring the global representations of the audios and videos $X_m^g$, $m \in \{a, v\}$, the Conv1D and transformer encoder are also adapted as in [33, 38]. Specifically, we process the representations of videos and audios $U_m$ by sequentially feeding them into a Conv1D layer, followed by transformer encoder layers to capture contextual information. We then apply maxpooling layer to aggregate a global representation from all tokens, as delineated in the subsequent expression:

$$X_m^g = \text{MAX}\left(\text{TE}\left(\text{Conv1D}\left(U_m\right)\right)\right) \quad (3)$$

here $MAX(\cdot)$ means the MaxPooling layer, $\text{TE}(\cdot)$ is short for Transformer Encoder, $m \in \{a, v\}$.

While for local representations of the audios and videos, we modify the approach by replacing the final maxpooling layer with adaptivemaxpooling to get $n$ local representations. This allows us to extract multiple local representations instead of a single global one. To maintain consistency across modalities, we utilize the same number of local representations, denoted as $n$, for all three modalities. The $n$ local representations of modality $m$ are finally concatenated,

denoted as $X_m^l$, $m \in \{a, v\}$. The equations can be seen as follows:

$$X_m^l = \text{CONCAT}\left(\text{AdaMaxPool}_n\left(\text{TE}\left(\text{Conv1D}\left(U_m\right)\right)\right)\right) \quad (4)$$

Thus, the global representations of the three modalities share the same dimension $X_m^g \in \mathbb{R}^d$, $m \in \{t, a, v\}$, so do the local representations $X_m^l \in \mathbb{R}^{nd}$, $m \in \{t, a, v\}$. Here $n$ is the hyperparameter to determine the number of the local representations of each modality in the given utterance.

### 3.4 Modality-Specific MoEs Module

Inspired by the successful application of sparse MoEs in areas such as bot detection [27] and large language models [10, 65], we capitalize on the principle of processing inputs through distinct experts followed by a cohesive combination. Recognizing the distinct value of each local representation in multimodal sentiment analysis and their variable importance across samples, we refine the combined local representation using a MoEs layer. This layer is crafted to selectively activate relevant experts, thus recognizing the importance of individual local representation and aiding in their combined analysis for a holistic understanding.

For simplicity, all modality-specific MoEs layers share the same number of the experts, denoted as $s$. Each modality-specific MoEs layer consists of a set of $s$ expert networks, $E_1, ..., E_s$, and a gating network $G$ with a sparse $s$-dimensional vector output [45]. The local representations $X_m^l \in \mathbb{R}^{nd}$, $m \in \{t, a, v\}$ are firstly fed into the

gate network $G$, which can be seen in:

$$G(X_m^l) = \text{Softmax}(\text{KeepTopK}(W_g X_m^l, \text{k})) \quad (5)$$

where $W_g \in \mathbb{R}^{nd \times s}$ denotes the learnable parameters, KeepTopK($\cdot$) denotes the function to choose the top $k$ higheset values given the input $X_m^l$. Then the outputs of all experts of the inputs $X_m^l$ are calculated as follows:

$$\hat{X}_m^l = \sum_{i=1}^{s} G(X_m^l)_i E_i(X_m^l) \quad (6)$$

where $G(X_m^l)_i$ denotes the possibility to assign local representations of modality $m$ to $i$-th expert and $E_i(X_m^l)$ denotes the output of the $i$-th expert. In another word, the most $k$ relevant experts will be selected to fuse and form the final refined local representation $\hat{X}_m^l \in \mathbb{R}^d$ using the weights from gate values.

In order to better regularize the MoEs layers and encourage all experts to have equal importance, following prior research [27, 45], for modality $m$, an additional loss $\mathcal{L}_{MoE}^m(X_m^l)$ is added:

$$\mathcal{L}_{MoE}^m(X_m^l) = \omega \cdot (\text{CV}(\text{Importance}(X_m^l))^2 + \text{CV}(\text{Load}(X_m^l))^2) \quad (7)$$

where CV($\cdot$) denotes the coefficient of variation, Importance($\cdot$) refers to the weighted importance scores of various expert networks, which is the output of gating network $G$, and Load($\cdot$) calculates the number of load samples currently present in each of the expert networks, which is defined in [45], $\omega$ is hyperparameter with default value $1e-2$.

## 3.5 Global-guided Fusion Module

Due to the 'few-to-many' tokens relation, local representations do not contain as much information as global presentations, and their predictive power is not the same [31]. Moreover, the contribution of different representations and modalities to the final outcome is not uniform; hence, it's vital to preserve the significance of each modality provided by global representations during fusion. To tackle these challenges, we have designed a global-guided fusion module that initially merges local and global representations within each modality, then blends all local and global representations to derive attention weights. The attention weights derived from the global feature fusion of the three modalities, are used as mixup weights to merge the modality-specific representations, thus safeguarding the significance of the original modalities in the composite representation.

Specifically, global and local representations corresponding to the three modalities are first stacked into a matrix $M \in \mathbb{R}^{6 \times d}$, then fed into the transformer encoder, which can be defined as follows:

$$\left[ Z_t^g, Z_a^g, Z_v^g, Z_t^l, Z_a^l, Z_v^l \right] = \text{Transformer Encoder}(M) \quad (8)$$

here denotes $Z_m^g \in \mathbb{R}^d$, $Z_m^l \in \mathbb{R}^d$ denotes the refined global and local representations of the modality $m$, $m \in \{t, a, v\}$, respectively.

These representations are then fused according to modality and the richness of the information they hold. Specifically, for modality $m$, a MLP layer is used to integrate the local and global representations to get the modality representation:

$$Z_m = \text{MLP}\left( \left[ Z_m^g, Z_m^l \right] \right) \in \mathbb{R}^d, m \in \{t, a, v\} \quad (9)$$

For global and local information, we apply attention functions [54] to individually merge three global representations and three local representations, obtaining the fused global representation $Z_g$ and fused local representation $Z_l$:

$$W_r, Z_r = \text{ATTN}\left( \left[ Z_t^r, Z_a^r, Z_v^r \right] \right) \in \mathbb{R}^d, r \in \{g, l\} \quad (10)$$

here $W_r$ denotes the attention weights of $r$ representations, and $Z_r$ denotes fused representations of $r$ representations, respectively.

After that, another MLP layer is used to fuse the $Z_g$ and $Z_l$:

$$Z_1 = \text{MLP}\left( \left[ Z_g, Z_l \right] \right) \in \mathbb{R}^d \quad (11)$$

For the modality representations, the mixup [28] is adapted to fuse these embeddings, with the weights from the $W_g$, which keeps the modality importance among all modalities:

$$Z_2 = \text{MIXUP}\left( (Z_t, Z_a, Z_v), W_g \right) \in \mathbb{R}^d \quad (12)$$

Finally, we use a MLP layer to get the prediction $\hat{y}$ based on the concatenation of the $Z_1$ and $Z_2$:

$$\hat{y} = \text{MLP}\left( [Z_1, Z_2] \right) \quad (13)$$

## 3.6 Optimization Object

The overall training of the GLoMo is perfomed by minimizing the following loss:

$$\mathcal{L} = \mathcal{L}_{task} + \mathcal{L}_{MoE}^t + \mathcal{L}_{MoE}^a + \mathcal{L}_{MoE}^v \quad (14)$$

here $\mathcal{L}_{task}$ involves regression and classification tasks. For regression task, following prior research [35, 39], the L1 loss, defined as the absolute difference between the predicted value $\hat{y}$ and the true label $y$, is used:

$$\mathcal{L}_{\text{reg}} = \frac{1}{n} \sum_{i=1}^{n} |y_i - \hat{y}_i| \quad (15)$$

while for classification, the standard cross-entropy loss is used:

$$\mathcal{L}_{\text{cla}} = \frac{1}{n} \sum_{i=1}^{n} -y_i log(\hat{y}_i) \quad (16)$$

where $n$ is the number of the samples.

# 4 EXPERIMENTS

## 4.1 Datasets

GLoMo is evaluated on multiple tasks, including multimodal sentiment analysis, multimodal humor detection and multimodal emotion recognition. The widely-used datasets, CMU-MOSI [58], CMU-MOSEI [59], CHERMA [47], UR-FUNNY [14] and MUStARD [6] are adapted. Due to the space limitation, the introduction of these datasets can be found in Appendix.

## 4.2 Evaluation Criteria

To comprehensively assess the performance of the proposed GLoMo, we adopt a set of widely-recognized metrics. The performance indicators utilized are as follows: (1) Accuracy-7 (Acc-7): This metric represents the accuracy over seven distinct sentiment intensity classes; (2) Binary Accuracy (Acc-2): This is the accuracy for binary classification tasks; (3) F1 Score: The F1 score is calculated for each sentiment category to provide a balance between precision and recall; (4) Mean Absolute Error (MAE): This represents the average

magnitude of errors in a set of predictions; (5) Correlation (Corr): This denotes the Pearson correlation between the predicted values and the ground-truth values. For the CMU-MOSI and CMU-MOSEI datasets, the Acc-2 and F1 score are reported in two forms using the segmentation marker '-/-': the former score is negative/non-negative socre while the latter one is the score for negative/positive. The difference between the non-negative and positive is that the former are based on scores $\geq 0$, while the latter containing scores $> 0$.

## 4.3 Feature Extraction

For fair comparison, we apply the same word-aligned embeddings as in [15, 38, 40].

**Text Features.** We utilize BERT [20], which has been established as the standard in prior research [15, 36, 40, 56]. Specifically, we employ the BERT [20] model for the CMU-MOSI, CMU-MOSEI, UR-FUNNY, and MUStARD datasets, and the Chinese BERT-base model [8] for the CHERMA.

**Audio Features.** We extract features such as Mel-frequency cepstral coefficients and pitch using COVAREP [9] for the MOSI, MOSEI, UR-FUNNY, and MUStARD datasets. For CHERMA, we use the pre-trained wav2vec model for feature extraction [62].

**Video Features.** We apply the Multitask Cascaded Convolutional Networks (MTCNN) [63] and OpenFace frameworks [3, 4] to detect faces and extract features such as facial action units, head pose, gaze orientation, and eye gaze from the CMU-MOSI, CMU-MOSEI, UR-FUNNY, MUStARD. For CHERMA datasets, we use MTCNN for face alignment and a pre-trained Resnet-18 model [16], which has been trained on the RAF-DB dataset, for feature extraction [21].

The extracted features vary in dimension across different datasets due to the diverse extraction methods and lengths of the utterances. For instance, in CMU-MOSI, the dimensions for text, acoustic, and visual features are 768, 74, and 47, respectively. In contrast, for CMU-MOSEI, they are 768, 74, and 35, respectively. The dimensions for UR-FUNNY and MUStARD are 768 for text, 81 for acoustic, and 91 for visual modalities. For CHERMA, the corresponding dimensions are 1024, 1024, and 2048.

## 4.4 Baselines

In our study, we have selected a variety of multimodal fusion methods as baselines to conduct a comprehensive comparison. These methods include models such as TFN [57], LMF [29], MFM [52], GFN [35], and ICCN [48], which directly fuse the global representations of the three modalities. Additionally, we consider approaches like the MULT [50] and BBFN [13], M3SA [60] algorithms, which first fuse pairs of global representations before integrating them together. We also explore methods like MISA [15], which partition the modalities' global representations into modality-specific and modality-common components. Furthermore, we investigate the importance of modality-specific tokens within each modality using algorithms such as Prisa [33], compare with CubeMLP [46] that use toekn-level fusion strategies and the state-of-the-art C-MIB [38] that use the mutual inforamtion for denoising.

## 4.5 Implementation Details

All experiments are conducted with the PyTorch framework on GTX3090 with CUDA 11.5 and torch version of 1.12.1. To ensure fair and consistent comparison, our proposed GLoMo is trained using AdamW [30] optimizer with a with a fixed random seed of 5576 [36, 40]. In all datasets, we consistently employed the same number of experts $s$, and the same quantity of local representations $n$. Specifically, all MoEs layers contained 3 experts, and the number of local representations for each modality was also set to 3. However, the learning rate and the hidden layer dimension were varied according to the dataset. For the MOSI dataset, the learning rate was set at $4e - 5$ with a hidden layer dimension of 48. In contrast, for the MOSEI dataset, the learning rate was reduced to $1e - 5$, and the hidden layer dimension was increased to 192. For the other datasets, such as UR-FUNNY, MUSTARD, and the remaining ones, the learning rate was standardized at $2e - 5$, but the hidden layer dimensions differed, being 112, 160, and 256 respectively. More details can be found in Appendix.

## 5 RESULTS AND ANALYSIS

## 5.1 Quantitative Results

**Multimodal Sentiment Analysis** Table 1 presents a comparison between our proposed GLoMo model and other baselines, with the best results highlighted in bold and the second-best underscored. As observed from Table 1, GLoMo surpasses all other models across all classification metrics on both datasets, achieving a new state-of-the-art. Specifically, GLoMo has improved by over 1% on the MOSI dataset for both Acc-2 and F1 positive metrics, even out-performing PRISA, which utilizes regression labels for supervised contrastive learning. This emphatically demonstrates the effectiveness of GLoMo's comprehensive utilization of global and local representations from various modalities, and underscores that local representations can enhance the provision of auxiliary information crucial for better emotion category recognition.

However, in terms of MAE and Corr, only PRISA outperforms GLoMo on both metrics across the datasets. This is attributed to PRISA's use of label information for supervised contrastive learning, which effectively increases the sample size. This highlights that without additional data, relying solely on global representations may result in lower MAE or higher correlation. The integration of local representations, on the other hand, can serve as a trade-off between these two aspects, elevating the model's performance to a desirable level.

**Multimodal Humor Detection and Multimodal Emotion Recognition** Tables 2 and 3 showcase the performance of the GLoMo model on the multimodal humor detection and multimodal emotion recognition tasks, respectively. Notably, GLoMo has out-performed the existing best models on all three benchmarks, setting a new state-of-the-art. Specifically, on the UR-FUNNY dataset, GLoMo achieved an improvement of 2.52%, and on the MUStARD dataset, it recorded a significant gain of 7.39%. Furthermore, GLoMo demonstrated superior performance in all seven distinct emotion categories on the CHERMA dataset, with an overall enhancement of 3.08% in the F1 score. This improvement may be attributed to the focus of these three datasets on detecting specific emotions, rather than the general sentiment polarity classification as seen

Table 1: Performances of multimodal models in MOSI and MOSEI. $\diamond$ from [61], ♠ from [64].

| Models | CMU-MOSI | | | | | CMU-MOSEI | | | | |
|---|---|---|---|---|---|---|---|---|---|---|
| | MAE (↓) | Corr (↑) | ACC-2 (↑) | F1(↑) | ACC-7 (↑) | MAE (↓) | Corr (↑) | ACC-2 (↑) | F1(↑) | ACC-7 (↑) |
| TFN [57]♠ | 0.901 | 0.698 | -/80.8 | -/80.7 | 34.9 | 0.593 | 0.700 | -/82.5 | -/82.1 | 50.2 |
| LMF [29]♠ | 0.917 | 0.695 | -/82.5 | -/82.4 | 33.2 | 0.623 | 0.677 | -/82.0 | -/82.1 | 48.0 |
| MFM [52]♠ | 0.877 | 0.706 | -/81.7 | -/81.6 | 35.4 | 0.568 | 0.717 | -/84.4 | -/84.3 | 51.3 |
| GFN [35]$\diamond$ | 0.736 | 0.790 | -/84.3 | -/84.3 | 47.0 | 0.611 | 0.774 | -/85.0 | -/85.0 | 51.8 |
| MULT [50]$\diamond$ | 0.767 | **0.799** | -/83.7 | -/83.7 | 41.5 | 0.625 | 0.775 | -/84.7 | -/84.6 | 50.7 |
| MAGBERT [44]$\diamond$ | 0.790 | 0.769 | -/83.5 | -/83.5 | 42.9 | 0.602 | 0.778 | -/85.0 | -/85.0 | 51.9 |
| M3SA [60]$\diamond$ | 0.730 | 0.793 | -/85.3 | -/85.3 | 45.5 | 0.599 | 0.781 | -/85.2 | -/85.1 | 52.5 |
| ICCN [48]$\diamond$ | 0.860 | 0.710 | -/83.0 | -/83.0 | 39.0 | 0.565 | 0.713 | -/84.2 | -/84.2 | 51.6 |
| CubeMLP [46] | 0.770 | 0.767 | -/85.6 | -/85.5 | 45.5 | 0.529 | 0.760 | -/85.1 | -/84.5 | 54.9 |
| MISA [15] | 0.783 | 0.761 | 81.8/83.4 | 81.7/83.6 | 42.3 | 0.555 | 0.756 | 83.6/85.5 | 83.8/85.3 | 52.2 |
| BBFN [13] | 0.776 | 0.755 | -/84.3 | -/84.3 | 45.0 | 0.529 | 0.767 | -/86.2 | -/86.1 | 54.8 |
| PriSA [33] | **0.714** | 0.792 | 83.4/85.5 | 83.2/85.5 | 47.3 | **0.523** | 0.772 | 82.8/85.9 | 83.2/85.9 | 54.7 |
| C-MIB [38] | 0.728 | 0.793 | -/85.2 | -/85.2 | 48.2 | 0.584 | **0.789** | -/86.2 | -/86.2 | 53.0 |
| GLoMo | 0.718 | 0.782 | **84.1/86.7** | **83.9/86.6** | **48.3** | 0.539 | 0.771 | **83.7/86.5** | **84.0/86.4** | **55.0** |

Table 2: The comparison with baselines on UR-FUNNY and MUStARD, in terms of ACC-2. Models in parentheses indicates the textual features used.

| | UR-FUNNY (↑) | MUStARD (↑) |
|---|---|---|
| MISA [15] (BERT) | 69.62 | 66.18 |
| MISA [15] (ALBERT) | 69.82 | 66.18 |
| MAGBERT [44] (ALBERT) | 67.20 | 69.12 |
| MAGBERT [44] (XLNet) | 72.43 | 76.47 |
| GLoMo (BERT) | **74.95** | **83.86** |

in CMU-MOSI and CMU-MOSEI. The detection of various specific emotions requires leveraging different local representations from each modality to aid in differentiation. GLoMo's utilization of multiple experts appears to effectively address this challenge, resulting in significant performance gains.

## 5.2 Ablation Study

In this section, we conduct ablation studies on the individual components of GLoMo, including the modalities, the application of MoEs, the roles of local and global representations, various fusion strategies, and the efficacy of global-fusion, as illustrated in Table 4.

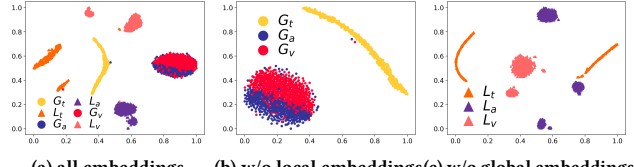

(a) all embeddings    (b) w/o local embeddings(c) w/o global embeddings

Figure 2: t-SNE plot of the global and local representations of the three modality of MOSI, where $G_m$ and $L_m$ denote the global and local representations of the modality $m$, $m \in \{t, a, v\}$

### 5.2.1 Role of Modalities.
As observed in Table 4, the performance of GLoMo diminishes to varying extents upon the removal of any modality, with a particularly notable decline of over 20% on both CMU-MOSI and CMU-MOSEI datasets when textual modality is excluded. This significant drop may be attributed to the dominant role that text plays in these datasets, while the representations of the other two modalities exhibit a considerable amount of overlap and redundancy, corroborating previous findings [34, 37, 64]. Consistent with prior studies [64], on the CMU-MOSI and CMU-MOSEI datasets, the performance degradation caused by omitting the visual modality is more pronounced than that caused by discarding the auditory modality. This could be due to the extensive redundancy between audio and visual representations, as depicted in Fig. 2, and the possibility that the visual modality encompasses more informative cues than the auditory modality. In contrast, the impact of removing a single modality is less severe in CHERMA, which might be linked to the fact that features from each modality are extracted using pretrained models, thus retaining a richer set of modality-specific information.

### 5.2.2 Role of MoEs.
In this section, we examined the role of the MoEs layer and the impact of varying the number of modality-specific experts. We introduced MLP_M, which utilizes one-layer modality-specific MLP in place of modality-specific MoEs layer for the fusion of local representations. As indicated in Table 4, substituting MoEs with MLPs led to a slight decline in the classification performance of GLoMo across three datasets; however, the performance remained superior to most models that rely solely on global representations as listed in Table 1 and 3. This underscores the effectiveness of incorporating local representations for enhanced feature representation. Furthermore, we investigated whether an increase in the number of experts in MoEs layer would enhance model performance. We conducted experiments with varying numbers of experts for text, visual, and audio modalities, set at 1, 2, 3, and 4, resulting in a total of 64 different configurations. The mean F1 scores for each modality and number of experts on the datasets are depicted in Fig. 3b, with detailed results available in

**Table 3: The comparison with baselines on CHERMA in terms of F1-score. The results are from [47].**

|  | Happiness (↑) | Sadness (↑) | Fear (↑) | Anger(↑) | Surprise (↑) | Disgust (↑) | Neutrality (↑) | overall (↑) |
|---|---|---|---|---|---|---|---|---|
| TFN [57] | 74.91 | 75.56 | 66.15 | 74.41 | 66.29 | 43.34 | 65.60 | 68.37 |
| LMF [29] | 74.52 | 75.83 | 66.73 | 74.55 | 65.08 | 45.70 | 65.64 | 68.23 |
| EFT [47] | 74.98 | 76.88 | 67.32 | 74.85 | 66.73 | 47.48 | 64.60 | 68.72 |
| LFT [47] | 75.07 | 76.29 | 66.80 | 74.88 | 66.67 | 47.74 | 65.97 | 69.05 |
| MULT [49] | 76.18 | 76.88 | 67.36 | 74.85 | 68.18 | 46.96 | 65.26 | 69.24 |
| PMR [32] | 75.68 | 76.46 | 67.97 | 75.43 | 67.37 | 48.93 | 66.59 | 69.53 |
| LFMIM [47] | 76.6 | 77.83 | 69.44 | 75.32 | 69.83 | 50.20 | 68.24 | 70.54 |
| GLoMo | **81.73** | **81.33** | **74.92** | **77.48** | **70.63** | **50.88** | **70.13** | **73.62** |

**Table 4: Ablation studies for modules in GLoMo on MOSI, MOSEI and CHERMA datasets. T=text, A=audio and V=video.**

| Configs | CMU-MOSI | | CMU-MOSEI | | CHERMA |
|---|---|---|---|---|---|
| | ACC-2(↑) | F1(↑) | ACC-2(↑) | F1(↑) | F1 (↑) |
| Role of Each Modality | | | | | |
| V+A | 55.2/55.9 | 54.9/55.7 | 69.6/65.5 | 66.3/61.0 | 71.03 |
| V+T | 82.3/84.9 | 82.1/84.8 | 82.5/86.0 | 83.0/86.0 | 67.48 |
| A+T | 82.5/84.1 | 82.5/84.2 | 81.3/85.4 | 81.9/85.5 | 68.72 |
| V+A+T | **84.1/86.7** | **83.9/86.6** | **83.7/86.5** | **84.0/86.4** | **73.62** |
| Role of MoEs | | | | | |
| MLP$_M$ | 83.1/85.7 | 82.9/85.6 | 83.2/85.9 | 83.5/85.8 | 72.91 |
| Role of Representations | | | | | |
| Global | 81.9/83.7 | 81.8/83.7 | 83.5/85.4 | 83.7/85.3 | 72.39 |
| Local | 82.3/83.7 | 82.3/83.6 | 81.5/85.8 | 82.0/85.8 | 72.37 |
| Role of Fusion | | | | | |
| SUM | 82.3/83.8 | 82.2/83.8 | 81.4/85.6 | 82.0/85.6 | 70.60 |
| CON | 83.2/84.9 | 83.2/84.9 | 82.3/85.8 | 82.8/85.8 | 72.60 |
| ATTN$_F$ | 82.6/85.5 | 82.4/85.3 | 82.1/85.8 | 82.6/85.7 | 73.21 |
| MUL | 81.5/82.8 | 81.5/82.8 | 82.0/85.6 | 82.5/85.5 | 72.27 |
| Role of Global-guided Fusion | | | | | |
| ATTN$_G$ | 82.9/85.3 | 82.7/85.2 | 83.2/85.9 | 83.5/85.8 | 72.89 |
| MLP$_G$ | 83.1/85.2 | 83.0/85.2 | 81.9/85.6 | 82.4/85.6 | 72.93 |
| ReAt$_G$ | 83.2/85.5 | 83.1/85.4 | 83.5/86.3 | 83.8/86.2 | 73.35 |

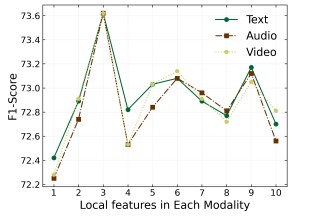 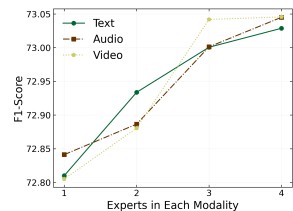

(a) **Number of Local Representations**     (b) **Number of Experts**

**Figure 3: Ablation studies on the number of local representations and experts on CHERMA.**

focus on different segments of information with little to no overlap. Additionally, global representations appear more concentrated, whereas local representations are dispersed into several clusters. This dispersion occurs because global representations are derived from max pooling across all tokens, whereas local representations are aggregated through multiple experts. Since we use three experts for each modality, focusing on different local representations, the local representations form three clusters of varying sizes. This clustering demonstrates the effectiveness of employing MoEs to concentrate on distinct local representations.

For a more intuitive understanding of the importance of global and local representations, we conduct ablation experiments, as shown in Table 4. Here, "Global" indicates the use of only global representations, and "Local" refers to the exclusive use of local representations. The results show that on the MOSI dataset, using only local representations yields better classification results. Conversely, on the MOSEI dataset, global representations perform better in classifications that include zero, while local representations excel in classifications excluding zero. On the CHERMA dataset, both feature types are equally effective. Furthermore, we explore whether an increase in the number of local representations correlates with improved model performance. As illustrated in Fig. 3a, having more local representations does not necessarily equate to better classification outcomes, as an excess of local representations can also introduce more redundant information.

*5.2.4 Role of Fusion.* In this module, we explore the impact of substituting the global-guided fusion module with alternative global and local fusion strategies, as shown in Table 4. The 'SUM' strategy involves a straightforward addition of all representations, while 'CON' refers to concatenating them before merging using a MLP.

Appendix. As observed from Fig. 3b, model performance improved with an increase in the number of experts for each modality. This improvement may be attributed to the task involving the classification of seven categories of emotions, where a greater number of experts can better capture the local representations necessary for various emotional states. When any one of the modalities had four experts, the mean F1 score exceeded 73, surpassing the performance achieved with MLPs. This finding highlights the MoE's superior capability in local representations.

*5.2.3 Role of Representations.* In this module, we analyze both global and local representations and their performance within the GLoMo framework. Moreover, we aim to investigate if an increase in the number of local representations across different modalities leads to an enhancement in model performance. Consistent with prior research [40], we map the global and local representations of each modality to a two-dimensional space and visualize them using t-SNE [53], as shown in Fig. 2. As observed in Fig. 2a, the global representations and local representations for each modality tend to

'ATTN$_F$' employs an attention mechanism to fuse the representations, and 'MUL' simply multiplies the representations together. It is evident that the 'SUM' and 'MUL' methods treat all representations as equally important, which can lead to the loss of diverse feature information, resulting in subpar performance across all three datasets. Conversely, the 'ATTN$_F$' and 'CON' strategies, despite being simple rather than sophisticated, demonstrate superior performance on all three datasets, even surpassing the majority of the baselines. This suggests the significance of local representations and the effectiveness of employing MoEs.

*5.2.5 Role of Global-guided Fusion.* In this module, we will analyze various variants of the global-guided fusion strategy. As shown in Table 4, 'ATTN$_G$' denotes the use of attention as a substitute for MIXUP, while 'MLP$_G$' indicates the replacement of MIXUP with a MLP layer, and 'ReAt$_G$' signifies the application of local-guided fusion, that is, employing the attention weights of local representations to perform MIXUP. It is evident that the performance of using 'MLP' to replace MIXUP is the least effective, even falling behind the 'ATTN$_F$' model, which further underscores the differing significance of global and local representations. The 'ReAt$_G$' strategy demonstrates only a slight decrease compared to the original GLoMo on the CMU-MOSEI dataset, and it also exhibits the best results among the three on the CMU-MOSI and CHERMA datasets. This suggests the guiding role of different local representations, particularly in text-dominant scenarios like MOSEI and MOSI, hinting at the importance of utilizing global-guided fusion during the integration process.

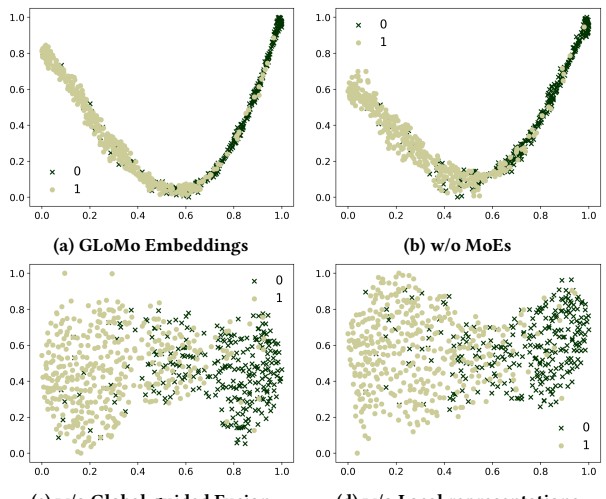

(a) GLoMo Embeddings

(b) w/o MoEs

(c) w/o Global-guided Fusion

(d) w/o Local representations

**Figure 4: t-SNE of the fused representations on CMU-MOSI, where '0' indicates samples with labels less than zero and '1' denotes samples with labels greater than zero.**

*5.2.6 Visualizing Representations.* In order to provide a more intuitive comparison of the representations obtained by GLoMo, we visualized the integrated representations using t-SNE, as shown in Fig. 4. Fig. 4a depicts the final representation acquired by GLoMo, Fig. 4b shows the representation using an MLP layer in place of

**Table 5: Comparison of parameters and running time for different models on the MOSI dataset.**

|  | # params ($\downarrow$) | running time ($\downarrow$) |
|---|---|---|
| MAGBERT [44] | 110,705,665 | 549s |
| MISA [15] | 110,620,273 | 535s |
| C_MIB [38] | 109,835,748 | 480s |
| TFN [57] | 161,409,399 | 275s |
| GLoMo | 109,818,887 | 481s |

MoEs layer, Fig. 4c illustrates the representation without the use of Global-guided fusion, and Fig. 4d presents the representation that excludes local representations. It is evident that replacing MoEs layer with MLP layer leads to a noticeable dispersion in the representations, suggesting that MoEs enhances the representational and discriminative capabilities, and indicating that the application of MoEs captures more relevant local information for the respective categories. The representations without the global-guided fusion module are particularly scattered, with a significant amount of overlap, demonstrating that global-guided fusion effectively guides the integration of global and local representations to achieve more distinctive representations. The representations excluding local representations are even more dispersed than those without global fusion, with increased overlap, implying that incorporating local representations enhances the discriminative power and category-specific representational ability.

*5.2.7 Complexity Analysis.* In this section, we measure the complexity of GLoMo. To provide a clear presentation and facilitate comparison, we perform a comparative analysis of the model's complexity, taking into account both the spatial and temporal dimensions. This analysis involves GLoMo and a variety of benchmark models on the MOSI dataset, the details of which are presented in Table 5. When evaluating spatial complexity, we use the number of parameters as the key metric. It is noted that GLoMo has the fewest parameters compared to the other methods. Regarding temporal complexity, we assess the running time of the models by conducting each for 100 epochs, which allows for a standardized comparison. Our findings indicate that GLoMo generally requires less time than both MAGBERT and MISA, and its running time is comparable to that of C_MIB. From the perspective of both spatial and temporal resource consumption, GLoMo demonstrates a lower number of parameters and reduced usage time, highlighting its lightweight nature and efficiency.

## 6 CONCLUSION

In this paper, we present GLoMo, a global-local modal fusion framework for multimodal sentiment analysis that integrates the multiple local representations and the global representations. GLoMo's tailored use of modality-specific experts finetunes local representations, while its innovative global-guided fusion module ensures a balanced integration, honoring the inherent 'few-to-many token' relationship. The framework's outstanding performance across diverse datasets and the validation provided by rigorous ablation studies highlight its robustness and the effective collaboration of its components.

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
