# OpenReview forum: "GLoMo: Global-Local Modal Fusion for Multimodal Sentiment Analysis"
_acmmm.org/ACMMM/2024/Conference — MM2024 Poster_

### Official Review · Reviewer_kZ4x · 2024-05-10

**Rating:** 2
**Confidence:** 3

**Summary:**

The paper considers the valuable role of local representations in each modality, which is overlooked in previous works, and develops a global-guided fusion module to address the "few-to-many" challenge caused when integrating a few local representations and one global representation. Experiments are carried out on several multimodal tasks and experimental results demonstrate the effectiveness of the proposed GLoMo methods.

**Strengths:**

The authors have elaborated their method in detail, and the paper is well-organized and easy to follow.
The addressed issues are novel and the technical sounds correctness.
Sufficient experiments and ablation studies are conducted to verify the proposed method and analyze their findings.

**Limitations:**

1. Although I appreciate the problems they are trying to solve, the proposed approach does not seem to be a good solution to these challenges. I look forward to seeing a better feature extraction method as this paper seems to claim the limitation caused by pooling operations, like Mean pooling. However, lots of simple Pooling operations are used in Section 3. It doesn't matter.
The more important issue is the way to obtain local representations. As described "These approaches overlook the local representations, which consist of fused representations of consecutive several tokens", each local representation as I understand it stands for consecutive token information. Using MoE to extract local representation is an available method when the inputs are token representations. However, in Equation (6), the input of each expert is the same vector $X^l_m$ and the output is a vector, what's the difference with MLP? The proposed method, in essence, is the stack of a lot of simple operations, like Pooling, MLP, ATTN, CONCAT, and CONV.  Finally, the method proposed does not lead to significant improvements as shown in Table 1. Although it seems to improve greatly in other tables, most of the selected compared methods are too old.

2. Although detailed descriptions of methods and ablation experiments, it does not significantly highlight the focus of the article. For example, the local representations are the core of this paper and I'm interested in Figure 2, but few valuable conclusions are drawn from Sections 5.2.1 and 5.2.3.
The other core lies in the global-guided fusion module, I suggest emphasizing the relationships between module structure and "few-to-many".

3. In Methodology, too many "undefined" functions are introduced, in my opinion, simplified well-known operations are OK, like FF, Conv1D, ATTN, MLP, and so on, but infrequent or custom operations need to be specified, I have no idea about how to process E(*) in Equation (6), Load(*) in Equation (7), AdaMaxPool(*), MIXUP(*) in Equation (12), and so on.
In addition, I suggest using obvious marks to distinguish different operations with the same abbreviation, like MLP, to represent they have different parameters.

4. Some other advice:
(1) Why select the last two layer tokens to generate local representations in Equation (2)? Maybe you should execute an ablation study.
(2) $s$ experts in Section 3.4, but $n$ in Figure 1.
(3) $W_g \in R^{(nd) \times s}$, what about $X^l_m$ that make Equation (5) is executable?

**Suitability:**

2

---

### Official Review · Reviewer_uTsM · 2024-05-12

**Rating:** 4
**Confidence:** 2

**Summary:**

In this paper, the authors propose GLoMo, a global-local modal fusion framework for multimodal sentiment analysis that integrates the multiple local representations and the global representations.The authors conduct extensive experiments. The framework’s outstanding performance across diverse datasets and the validation provided by rigorous ablation studies highlight its robustness and the effective collaboration of its components.

**Strengths:**

1.This paper carries out sufficient experiments to verify the effectiveness of the proposed framework.
2.The structure of GLoMo is very concise, but it is effective.

**Limitations:**

1.The Global-guided fusion module in Figure 1 is perplexing, it is suggested to present it in a clearer way.
2.The format is not standard enough, punctuation needs to be added after the formula.

**Suitability:**

3

---

### Official Review · Reviewer_izAF · 2024-05-14

**Rating:** 4
**Confidence:** 3

**Summary:**

This article centers on the Multimodal Sentiment Analysis task and introduces the GlobalLocal Modal (GLoMo) Fusion framework. GLoMo consists of two key parts: (i) modality-specific mixture of experts layers that blend various local representations within each modality, and (ii) a global-guided fusion module that efficiently merges global and local representations to handle the 'few-to-many token' relationship adeptly.

**Strengths:**

1. This work ran many experiments on various benchmarks/datasets, and the results are promising and competitive.

2. This paper is well-written, easy to read, and detailed. The figures are well-designed.

**Limitations:**

1. The aspect of valuable visualization presenting in local representations is not too problematic in these problems because usually, the samples are not too long and complicated that global representations can miss important tokens. Can you give some case studies/examples to show these issues clearly?

2. What is the contribution of MoEs in the Modality-Specific MoEs Module that is different from other MoEs being applied in other problems? (scale up the model or dataset size with the same computation cost)

3. The visualization should be labeled in the regression mode since the annotations of MOSI (MOSEI) are continuous and range from -3 to 3. With continuous visualization, it can show more clearly how the approaches work.

**Suitability:**

3

---

### Official Review · Reviewer_H3Cp · 2024-05-15

**Rating:** 3
**Confidence:** 3

**Summary:**

Multimodal Sentiment Analysis (MSA) has witnessed remarkable progress and gained increasing attention in recent decades. However, the integration of multiple local representations and the fusion of local and global information present significant challenges. Therefore, this paper proposes the Global-Local Modal (GLoMo) Fusion framework for MSA task.

**Strengths:**

This  proposed Global Local Modal (GLoMo) Fusion framework, which consists of two basic components: one is modality-specific mixture of expert layers that integrate diverse local representations within each modality, and a global-guided fusion module that effectively
combine global and local representations. The idea of the article is relatively novel.

**Limitations:**

(1)The experimental results have not improved greatly than other baselines.
(2)There are few pictures in the article, and it is suggested that more diagrams or algorithm pseudocodes can be added to show the main steps and processes of the method more intuitively.
(3)There is an issue with the formatting of the paper, as the tables is too far from the description in the main text, such as Table 3. It is not convenient for reading.
(4)Grammatical and spelling errors: 23 lines of "combine" should be replaced by "combines".

**Suitability:**

3

---

### Meta-Review · Area_Chair_b7P7 · 2024-07-03

**Recommendation:** Accept (Poster)
**Confidence:** 5

**Metareview:**

Test-Time Adaptation for Multimodal Sentiment Analysis